# Nutritional intakes of highly trained adolescent swimmers before, during, and after a national lockdown in the COVID-19 pandemic

**Josh W. Newbury**[1], **Wee Lun Foo**[2], **Matthew Cole**[1], **Adam L. Kelly**[1], **Richard J. Chessor**[3], **S. Andy Sparks**[4]*, **Mark A. Faghy**[5], **Hannah C. Gough**[6], **Lewis A. Gough**[1]

1 Department of Sport and Exercise, Human Performance and Health Research Group, Centre for Life and Sport Sciences (CLaSS), Birmingham City University, Birmingham, United Kingdom, 2 Research Institute for Sport and Exercise Sciences, Liverpool John Moores University, Liverpool, United Kingdom, 3 Sport Science and Sport Medicine Team, British Swimming, Loughborough, Leicestershire, United Kingdom, 4 Department of Sport and Physical Activity, Sports Nutrition and Performance Research Group, Edge Hill University, Ormskirk, United Kingdom, 5 School of Science, Sport and Exercise, University of Derby, Derby, United Kingdom, 6 TP Health, Towcester, Northamptonshire, United Kingdom

* andy.sparks@edgehill.ac.uk

**Data Availability Statement:** All relevant data are within the paper and its Supporting Information files.

## Abstract

Strict lockdown measures were introduced in response to the COVID-19 pandemic, which caused mass disruption to adolescent swimmers' daily routines. To measure how lockdown impacted nutritional practices in this cohort, three-day photograph food diaries were analysed at three time points: before (January), during (April), and after (September) the first UK lockdown. Thirteen swimmers (aged 15 ± 1 years) from a high-performance swimming club submitted satisfactory food diaries at all time points. During lockdown, lower amounts of energy (45.3 ± 9.8 vs. 31.1 ± 7.7 kcal·kg BM·day$^{-1}$, $p<0.001$), carbohydrate (5.4 ± 1.2 vs. 3.5 ± 1.1 g·kg BM·day$^{-1}$, $p<0.001$), protein (2.3 ± 0.4 vs. 1.7 ± 0.4 g·kg BM·day$^{-1}$, $p = 0.002$), and fat (1.6 ± 0.4 vs. 1.1 ± 0.3 g·kg BM·day$^{-1}$, $p = 0.011$) were reported. After lockdown, no nutritional differences were found in comparison compared to before lockdown (energy: 44.0 ± 12.1 kcal·kg BM·day$^{-1}$; carbohydrate: 5.4 ± 1.4 g·kg BM·day$^{-1}$; protein: 2.1 ± 0.6 g·kg BM·day$^{-1}$; fat: 1.5 ± 0.6 g ·kg BM·day$^{-1}$, all $p>0.05$), despite fewer training hours being completed (15.0 ± 1.4 vs. 19.1 ± 2.2 h·week$^{-1}$, $p<0.001$). These findings highlight the ability of adolescent swimmers to alter their nutrition based on their changing training circumstances when receiving sport nutrition support. However, some individuals displayed signs of suboptimal nutrition during lockdown that were not corrected once training resumed. This warrants future research to develop interactive education workshops that maintain focus and motivation towards optimal nutrition practices in isolated periods away from training.

## Introduction

In late 2019, a novel strain of the coronavirus SARS-CoV-2 was identified. This virus is the cause of the disease COVID-19, which has the potential to produce severe respiratory distress

**Funding:** The authors received no specific funding for this work.

**Competing interests:** The authors have declared that no competing interests exist.

within infected individuals [1]. The disease was rapidly spread worldwide through aerosols and droplets that are released when breathing, speaking, or coughing [2], leading to over 210 million cases and 4.4 million deaths as of August 2021 [3]. In an attempt to slow the spread of COVID-19, the UK Government introduced a national lockdown in March 2020. Under these conditions, all non-essential travel and activities were prohibited, which caused the cancellation or postponement of sporting events at the local (e.g., school sports), national (British Championships), and international level (Olympic Games) [4]. Moreover, all school and indoor leisure facilities were forced to close, having a large impact on the capacity for adolescent swimmers (aged ≤19 years) to complete their normal training programmes [5].

Adolescents from high-performance swimming clubs typically engage in 8–10 pool-based training sessions·week$^{-1}$, covering between 40–60 km·week$^{-1}$ depending on their specialist events [6, 7]. In addition, most swimmers also complete 3–5 land-based strength and conditioning sessions·week$^{-1}$, equating to a total weekly training time of over 15 hours·week$^{-1}$ under normal circumstances [6, 7]. Since most swimming clubs train at public, indoor leisure facilities, the imposed lockdown therefore had a significant impact on the quality and quantity of training sessions that could take place. For example, during the initial lockdown (April 2020), South African athletes (n = 1080, swimmers = 30) reported that training mostly consisted of land-based cardiovascular and bodyweight strength exercises, completed at moderate intensities for between 30–60 min·day$^{-1}$ [8]. Considering that these average training demands are greatly reduced compared to the standard training for high-performance swimmers, it is plausible that to suggest that losses in muscle mass and technical conditioning are likely to occur when lockdown conditions exceed 2–4 weeks [9]. Large reductions in training hours are also likely to decrease daily energy expenditure (DEE), which could result in body fat accretion if normal recommended dietary practices (i.e., energy: 35–75 kcal·kg BM·day$^{-1}$ [7], carbohydrate (CHO): 3–8 g·kg BM·day$^{-1}$ [7], fat: 1–2 g·kg BM·day$^{-1}$ [10]) are not reduced to meet the new training demands [11]. Each of these adaptations may have adverse implications for swimming performance, and it is imperative that highly trained adolescent swimmers maintain a nutritional intake that matches the new training demands during extended periods away from normal training.

Studies in rugby players and para-athletes have found that dietary intakes were either maintained or increased during the COVID-19 pandemic, despite an overall reduction in training volume [12, 13]. However, whether the same dietary behaviours would be replicated by adolescent swimmers is unclear, especially since this population often demonstrates body image concerns [14–17]. Based on this notion, it is likely that some adolescent swimmers turned to contemporary low-energy diets (e.g., low-fat, ketogenic, intermittent fasting [18]) in order to preserve body composition when physical activity was reduced. This is problematic as such dietary approaches could lead to immunosuppression if not managed correctly [19, 20], therefore increasing the risk of contracting respiratory infections. Indeed, the dietary intakes of athletes and the general population were shown to have declined in quality during lockdown periods [8, 21]. The combination of these factors could also result in the occurrence of suboptimal protein (<1.5–1.7 g·kg BM·day$^{-1}$ [7]) and micronutrient intakes [22] being consumed, which is a further cause for concern as these can increase the rate of muscle atrophy [9] and lead to deficiencies [23], respectively. To maintain health and performance, it is therefore clear that adolescent swimmers would benefit from sport nutrition support and education during times of isolation and restricted training. There is, however, a lack of research in adolescent swimmers to accurately inform these provisions, and this is compounded by the possible nutritional changes during and after a national lockdown being unknown. Hence, the purpose of this study was to retrospectively analyse the nutritional intakes of highly trained adolescent swimmers on three occasions: (a) before COVID-19 (January 2020), (b) during a national

**Table 1. Characteristics of the study participants.**

|  | Combined (n = 13) | Male (n = 5) | Female (n = 8) |
|---|---|---|---|
| Age (years) | 15 ± 1 | 15 ± 2 | 16 ± 1 |
| Body mass (kg) | 58.4 ± 8.5 | 55.6 ± 12.2 | 60.2 ± 5.5 |
| Height (m) | 1.66 ± 0.09 | 1.67 ± 0.02 | 1.66 ± 0.07 |
| Time competitive (years) | 5.6 ± 1.6 | 4.8 ± 1.8 | 6.3 ± 1.0 |

Mean ± standard deviation.

lockdown (April 2020), and (c) after lockdown regulations had eased (September 2020) to understand the dietary practices that occurred within this population.

## Materials and methods

This research took place within a British, high-performance, amateur swimming club whereby 26 swimmers initially met the research criteria (aged 13–19 years, nationally competitive). Based on this study's retrospective analysis of food diaries, 10 swimmers were excluded from data analyses due to unsatisfactory contributions, whereas three swimmers ended their participation within the sport before the study ended. Thirteen participants (Table 1) were therefore utilised due to research constraints [24]. At the time of the study, seven participants (54%) were ranked within the top ten swimmers in Great Britain for their respected age group in at least one event, whilst six (46%) had qualified for Olympic selection trials for the 2020 Games. Ethical approval was granted for this study by Birmingham City University (Newbury/7594/R (B)/2020/Aug/HELS FAEC) and both the swimmers and their parents/guardians provided written informed consent prior to participation.

### Protocol and measurements

All swimmers completed a mobile-based photograph assessment food diary in line with previously validated methods in highly trained adolescents [25]. Over a 72-hour period, including two training days and one rest day, each swimmer took two photographs of every food and fluid item consumed. The first photograph displayed a clear image of food and drink before the meal was started. The second showed the amount of food and drink remaining after the meal was finished. If the items were consumed in their entirety, then a photograph was still required as confirmation. To standardise food and drink portions, participants were provided with a paper 1x1 cm grid placemat to include in each photograph, and measurement shakers were used for all poured fluids [26]. Photographs were sent immediately to the lead researcher in real-time using a cellular, picture messaging smartphone application (WhatsApp, Mountain View, CA, USA). Further details were also given regarding the brand of product, cooking methods, and a clear description of the items in each meal using either text or voice recordings. For unsatisfactory contributions, participants were immediately contacted asking for further clarification.

### Nutritional information

The images and details of nutritional intake were collected and inputted into dietary analysis software (Nutritics 3.06, Dublin, Ireland) for each participant by the lead researcher. Any food items that were not available within the Nutritics database were manually entered using information from the packaging label. This provided information regarding the energy and macro-nutrient content of each food and drink item consumed, as well as further details on the CHO

(e.g., sugar, fibre) and lipid components (e.g., omega-3 fatty acids, saturated fat), minerals (e.g., calcium, iron), trace elements (e.g., copper, zinc), vitamins (e.g., vitamin D, vitamin B12), and caffeine intakes consumed on each day that food diaries were submitted. The mean of the three days was used to estimate the typical dietary intakes of the swimmers over the three sampling time points as per previous recommendations [25]. All nutrients were reported in absolute values, with energy, macronutrients, and caffeine further discussed as relative values to standardise intakes based on body mass.

A small pilot study was conducted beforehand to assess the sensitivity of the analysis, which involved the lead researcher analysing eight meals with known energy and macronutrient content on two separate occasions. Additionally, one of the 600 kcal meals was repeatedly analysed on 20 separate occasions. These data were used to determine the intra-rater reliability and the validity of the assessment method of the lead researcher. Intra-rater reliability was assessed using a coefficient of variation (CV) and was determined to be 2.7%, 4.8%, 3.9% and 1.2% for energy (MJ), carbohydrate (CHO), protein and fat (g) respectively. The validity of the analysis method was assessed using Pearson's correlation coefficient and by the CV of error. For energy, CHO, protein, and fat, these were calculated as $r = 0.947$, $p<0.001$, CV = 2.0%; $r = 0.972$, $p<0.001$, CV = 3.5%; $r = 0.914$, $p<0.001$, CV = 6.4%; and $r = 0.977$, $p<0.001$, CV = 1.0%, respectively.

## Anthropometric data

Body mass was determined by the use of electronic scales (Seca 813, Hamburg, Germany) both before and after lockdown. All measurements were collected before morning training (4:45 AM) whilst the participants were in swimming costumes. Due to social distancing guidelines during lockdown, participants provided their body mass remotely using their own home-based electronic scales. In an attempt to standardise measures, these were requested after breakfast whilst wearing underwear. These participants' scales varied in accuracy; however, this was an unavoidable circumstance given the restrictions. Height was also measured before and after lockdown using a stadiometer (Seca 213, Hamburg, Germany), although this could not be measured during lockdown since swimmers did not have access to this equipment at home. All measures after lockdown were collected in line with the close contact guidelines set by the UK Government [27].

## Training information

Weekly swimming volume was defined as the cumulative number of kilometres covered by swimming each week. This information was collected for each participant via communication with the head swimming coach, who kept records of all swimming sessions run by the club before and after lockdown. During lockdown, some participants were able to complete some swimming volume in open water (n = 6) or private swimming facilities (n = 2), in which the swimming coach recommended session intensities and volumes. Actual swimming volumes were self-reported following the session. Weekly training time accounted for the total hours spent in exercise (pool and land-based) each week. Before lockdown, training consisted of between 5–8 pool (1.5–2.5 hours) and 2–6 gym-based (45–60 min) sessions·week[-1] based upon training age and specialist distance. After lockdown, training consisted of between 5–7 pool (1–1.5 hours) and 2–3 gym-based (60 min) sessions·week[-1]. The swimmers were asked to detail any additional exercise outside of these scheduled sessions during food diary collection. During lockdown, training was self-governed by the swimmers based upon recommendations made by coaches and support staff. Each swimmer self-reported their weekly training schedules to coaches and support staff during this time.

## Nutrition support

In January 2020, all participants completed three-day food diaries as part of routine nutrition support provided by the lead researcher and an undergraduate student. The obtained information was used to provide each swimmer with individual feedback (face-to-face, interventions), whereas group trends were used as topics for classroom-based education sessions (varying topics, 30 min, weekly). The level and frequency of individual support was determined by the swimmer's engagement. In immediate response to the lockdown, nutrition resources (PDF files) were created and sent to swimmers and their parents (WhatsApp, Mountain View, CA, USA) on ad hoc basis in March and April. These mostly included topics centred around training from home, muscle retention, spontaneously reducing energy intake, and improving immunity. The April food diary was requested to monitor adherence to the resources and identify possible nutritional interventions. Each swimmer received a phone or video call (WhatsApp) from the lead researcher to discuss their results, with the further option of weekly check-ins until training resumed. The level of support was again determined by the swimmer's engagement with the service. From April until June, the undergraduate student provided two group nutrition activities per week (cooking workshop, nutrition quiz) using an online meeting platform (Zoom, San Jose, CA, USA). Attendance at these activities was not monitored. All online group activities ceased in July as the swimmers began to resume formal training. The weekly check-in option remained in place since face-to-face contact was not prohibited. Resources for pre-, intra-, and post-training nutrition (e.g., nutrient timing, hydration, CHO) were sent to swimmers and parents to facilitate the return to training throughout August 2020.

## Statistical analyses

Descriptive and statistical analyses were undertaken using SPSS (version 25; IBM, New York, NY, USA). Normality of all data was verified by using visual inspection of Q-Q plot, histogram, and Shapiro-Wilk statistics. A one-way ANOVA was used to establish mean differences for nutritional intake variables and body mass at the three sampling time points (before, during, after lockdown). If the sphericity of the data was violated (checked via Mauchly test), then the appropriate Huyn-Feldt (epsilon value >0.75) or Greenhouse-Geiser (epsilon value <0.75) corrections were applied. Statistical significance was set at $p \leq 0.05$ and effect sizes were reported as partial eta squared ($P\eta^2$). Differences that met this threshold for statistical significance were subject to post hoc comparisons (Bonferroni), with effect sizes for comparisons between two time points calculated using the Hedge's $g$ bias correction based on the small sample size (n<20) of the study [28]. Effect sizes were interpreted for $P\eta^2$ (small: 0.01–0.05, medium: 0.06–0.13, large: >0.13) and $g$ (small: 0.20–0.49, medium: 0.50–0.79, large: >0.79) in accordance with Cohen [29]. A paired samples t-test was used to determine differences in height before and after lockdown. All data were reported as mean ± standard deviation (SD). Coefficient of variation (CV) was calculated and reported using SD/mean*100.

Statistics for the pilot tests utilised a two-way mixed model intra-class correlation co-efficient for reliability analysis (type: absolute agreement) and are reported with $r$ value and significance level. Interpretation of reproducibility was determined by the respective $r$ value with categories of poor ($\leq$0.39), fair (0.40–0.59), good (0.60–0.74), and excellent ($\geq$0.75) [30].

## Results

### Training characteristics

The COVID-19 pandemic caused significant disruptions to swimming volume (F = 139.8, $p<0.001$, $P\eta^2 = 0.92$), such that a 91% reduction occurred from before (43.3 ± 11.7 km·week$^{-1}$)

to during lockdown (4.1 ± 5.8 km·week$^{-1}$; $p<0.001$, $g = 4.11$). Furthermore, social restrictions in place after lockdown meant that swimming volume did not fully return once formal training resumed (33.5 ± 7.0 km·week$^{-1}$; vs. before lockdown: $p<0.001$, $g = 0.98$). The same pattern was observed for training time (F = 187.8, $p<0.001$, Pη$^2$ = 0.94), whereby swimmers reduced their time exercising by 70% during lockdown (19.1 ± 2.2 vs. 5.7 ± 2.5 h·week$^{-1}$, $p<0.001$, $g = 5.51$). The return of swimming training after lockdown did not result in a complete return of training hours compared to before lockdown (15.0 ± 1.4 h·week$^{-1}$; $p<0.001$, $g = 2.15$).

## Anthropometric responses

Body mass increased over the course of the study (F = 30.9, $p<0.001$, Pη$^2$ = 0.72), although this change did not occur during the initial stages of lockdown (58.4 ± 8.5 vs. 59.0 ± 8.6 kg, $p = 0.254$, $g = 0.07$). After lockdown, however, body mass was increased compared to measurements taken both before and during lockdown (62.7 ± 9.4 kg, both $p<0.001$, $g = 0.46$ and 0.40, respectively). This increase in body mass was accompanied by an increased height recorded from before to after lockdown (1.66 ± 0.09 vs. 1.69 ± 0.08 m, $p = 0.002$, $g = 0.27$).

## Energy and macronutrient intakes

Changes in absolute and relative dietary intakes of energy, CHO, protein, and fat were observed over the study timeframe (Table 2). The most prominent of these changes occurred during lockdown, whereby reductions in relative energy (-14.2 ± 8.2 kcal·kg BM·day$^{-1}$, $p<0.001$, $g = 1.56$), CHO (-1.9 ± 1.1 g·kg BM·day$^{-1}$, $p<0.001$, $g = 1.60$), protein (-0.6 ± 0.5 g·kg BM·day$^{-1}$, $p = 0.002$, $g = 1.45$), and fat intakes (-0.5 ± 0.4 g·kg BM·day$^{-1}$, $p = 0.011$, $g = 1.37$) were observed compared to before lockdown values. After lockdown, relative energy (-12.9 ± 10.5 kcal·kg BM·day$^{-1}$, $p = 0.003$, $g = 1.23$), CHO (+1.9 ± 1.1 g·kg BM·day$^{-1}$, $p<0.001$, $g = 1.46$), and protein (+0.4 ± 0.6 g·kg BM·day$^{-1}$, $p = 0.002$, $g = 1.45$) intakes increased again to match the level of consumption observed before lockdown. Relative fat intakes did not increase between these two time points ($p = 0.106$), despite a large effect size being calculated (+0.4 ± 0.6 g·kg BM·day$^{-1}$, $g = 0.82$). This was likely caused by an increased inter-individual variance at this time (CV = 36%, increased: n = 8, maintained: n = 4, decreased: n = 1) which

**Table 2. Energy and macronutrient intakes of highly trained adolescent swimmers recorded before, during, and after the COVID-19 lockdown.**

| Nutrition Variable | Lockdown Period | | | Interaction effect |
|---|---|---|---|---|
| | **Before** | **During** | **After** | |
| **Energy Intake** | | | | |
| Absolute (kcal·day$^{-1}$) | 2606 ± 507 | 1796 ± 338[a] | 2712 ± 718[b] | F = 17.5, $p<0.001$, Pη$^2$ = 0.60 |
| Relative (kcal·kg BM·day$^{-1}$) | 45.3 ± 9.8 | 31.1 ± 7.7[a] | 44.0 ± 12.1[b] | F = 17.9, $p<0.001$, Pη$^2$ = 0.60 |
| **Carbohydrate Intake** | | | | |
| Absolute (g·day$^{-1}$) | 311.1 ± 67.7 | 200.1 ± 41.0[a] | 329.7 ± 77.3[b] | F = 26.4, $p<0.001$, Pη$^2$ = 0.70 |
| Relative (g·kg BM·day$^{-1}$) | 5.4 ± 1.2 | 3.5 ± 1.1[a] | 5.4 ± 1.4[b] | F = 27.5, $p<0.001$, Pη$^2$ = 0.70 |
| **Protein Intake** | | | | |
| Absolute (g·day$^{-1}$) | 134.6 ± 25.9 | 98.9 ± 24.2[a] | 132.3 ± 38.0[b] | F = 8.3, $p = 0.002$, Pη$^2$ = 0.41 |
| Relative (g·kg BM·day$^{-1}$) | 2.3 ± 0.4 | 1.7 ± 0.4[a] | 2.1 ± 0.6[b] | F = 8.4, $p = 0.002$, Pη$^2$ = 0.41 |
| **Fat Intake** | | | | |
| Absolute (g·day$^{-1}$) | 91.6 ± 21.8 | 66.7 ± 17.6[a] | 95.1 ± 34.3 | F = 6.2, $p = 0.007$, Pη$^2$ = 0.34 |
| Relative (g·kg BM·day$^{-1}$) | 1.6 ± 0.4 | 1.1 ± 0.3[a] | 1.5 ± 0.6 | F = 5.7, $p = 0.009$, Pη$^2$ = 0.32 |

[a] = different to before lockdown ($p \leq 0.05$)

[b] = different to after lockdown ($p \leq 0.05$). Mean ± standard deviation.

has higher compared to both CHO (CV = 27%, increased: n = 13) and protein (CV = 28%, increased: n = 11, decreased: n = 2).

Indeed, similar individual patterns emerged for all relative energy and macronutrient intakes across the sampling timeframe, such that reductions occurred during lockdown, before intakes increased again albeit with a larger inter-individual variance (before vs. after CVs; energy: 22% vs. 28%, CHO: 22% vs. 27%, protein: 19% vs. 28%, fat: 28% vs. 36%; Fig 1). This subsequently affected how many swimmers were achieving the nutritional recommendations at each time point. Most notably, this resulted in an increase in swimmers falling below the minimal recommended intakes for protein (before: n = 0, after: n = 1) and fat (before n = 0, after: n = 2). Moreover, whereas the same number of swimmers (n = 3) were intaking $<35$ kcal·kg BM·day$^{-1}$ of energy at both time points, the after lockdown values shown that these

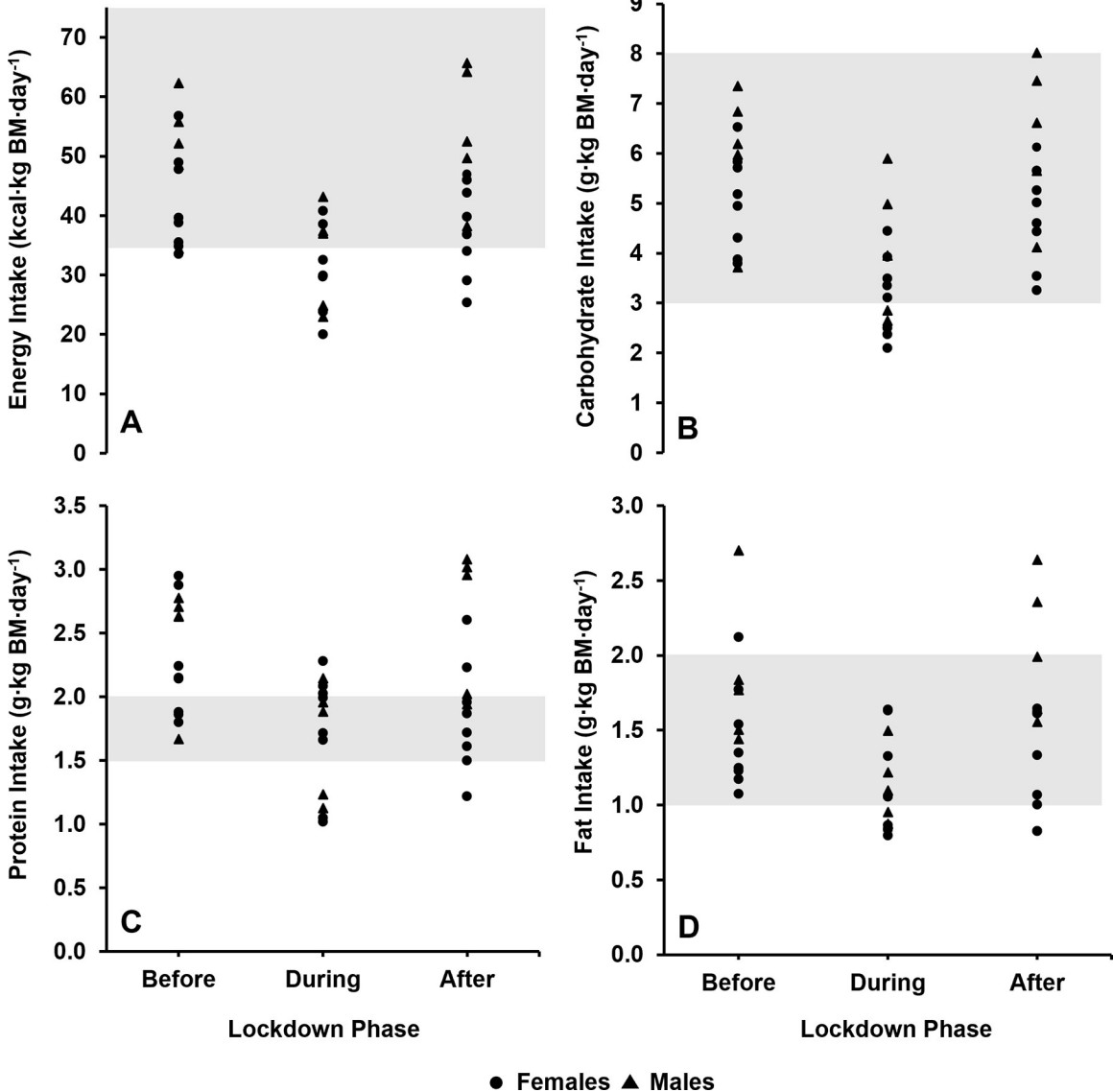

● Females ▲ Males

**Fig 1. The relative energy and macronutrient intakes of each individual swimmer before, during, and after the COVID-19 lockdown.** Grey shaded areas represent the recommended nutritional intakes for swimmers based upon: A) energy intake [7], B) CHO intake [7], C) protein intake [7], D) fat intake [10].

swimmers' relative intakes were further from the recommended threshold (before: 33.5, 34.5, and 34.8 kcal·kg BM·day$^{-1}$ vs. after: 25.3, 29.0 and 34.0 kcal·kg BM·day$^{-1}$). In contrast, more swimmers began to consume protein within the recommended range after lockdown (before: n = 4, after: n = 7) compared to the majority of swimmers (n = 9) intaking >2 g·kg BM·day$^{-1}$ before the lockdown was issued.

## Carbohydrate and lipid components

Changes in dietary fibre (F = 13.4, $p < 0.001$, P$\eta^2$ = 0.53), sugar (F = 12.6, $p < 0.001$, P$\eta^2$ = 0.51), and monounsaturated fat (F = 5.8, $p$ = 0.009, P$\eta^2$ = 0.33) intakes were identified over the study timeframe. Before lockdown, 12 swimmers were consuming adequate dietary fibre (25 g [31]). However, during lockdown, there was a 30% reduction in fibre intake ($p$ = 0.001, g = 1.87) that resulted in 10 swimmers falling below the daily recommendation. After lockdown, mean fibre intake returned to before lockdown levels ($p$ = 1.000, g = 0.16), although more swimmers (n = 4) were still intaking inadequate amounts. Sugar intakes followed a similar pattern, whereby a 37% reduction occurred from before to during lockdown ($p$ = 0.038, g = 0.88), and returned to similar levels after lockdown had ended ($p$ = 0.223, g = 0.41). All swimmers consumed sugar above the daily recommendation (≤5% of total energy intake [31]) at all time points, but this was highest after lockdown (before: 18 ± 6%, during: 16 ± 6%, after: 21 ± 6%). Monounsaturated fat intake also decreased over the study timeframe, with this change becoming apparent after lockdown (vs. before: -28%, $p$ = 0.005, g = 0.85). A large effect size suggests that a decrease also occurred from before to during lockdown (g = 1.12), although this change did not reach statistical significance ($p$ = 0.068). Saturated fat intakes were close to the recommendation (≤11% of total energy intake [31]) at all three time points (before: 11 ± 2%, during: 12 ± 3%, after: 11 ± 2%). No other carbohydrate or lipid component changed considerably over the study timeframe (Table 3).

## Minerals and trace elements

The dietary intake of 7 out of the 12 measured minerals and trace elements reduced during the lockdown period (Table 4), with sodium ($p$ = 0.010, g = 1.32) potassium ($p$ = 0.030, g = 1.14), chloride ($p$ = 0.010, g = 1.39), calcium ($p$ = 0.009, g = 0.99), phosphorus ($p$ = 0.002, g = 1.34), iron ($p$ = 0.004, g = 1.33), and selenium ($p$ = 0.007, g = 1.00) all being affected. After the lockdown, however, only sodium ($p$ = 0.036, g = 1.05), chloride ($p$ = 0.009, g = 1.36), and calcium

**Table 3. Carbohydrate and lipid components recorded in the diets of highly trained adolescent swimmers before, during, and after the COVID-19 lockdown.**

| Nutrient | Lockdown Period | | | Interaction Effect |
|---|---|---|---|---|
| | **Before** | **During** | **After** | |
| **Fibre (g·day$^{-1}$)** | 29.7 ± 4.3 | 20.7 ± 5.0$^a$ | 30.8 ± 8.1$^b$ | F = 13.4, $p < 0.001$, P$\eta^2$ = 0.53 |
| **Sugars (g·day$^{-1}$)** | 118.1 ± 57.2 | 74.5 ± 36.1$^a$ | 139.4 ± 41.1$^b$ | F = 12.6, $p < 0.001$, P$\eta^2$ = 0.51 |
| **Saturated Fat (g·day$^{-1}$)** | 32.3 ± 8.9 | 24.8 ± 8.3 | 33.3 ± 12.9 | F = 2.9, $p$ = 0.069, P$\eta^2$ = 0.20 |
| **Monounsaturated Fat (g·day$^{-1}$)** | 28.6 ± 9.7 | 20.6 ± 6.1 | 20.5 ± 8.8$^a$ | F = 5.8, $p$ = 0.009, P$\eta^2$ = 0.33 |
| **Polyunsaturated Fat (g·day$^{-1}$)** | 12.1 ± 4.2 | 9.6 ± 3.6 | 9.8 ± 3.3 | F = 2.8, $p$ = 0.081, P$\eta^2$ = 0.19 |
| **Omega-3 Fatty Acids (g·day$^{-1}$)** | 2.0 ± 1.2 | 1.7 ± 1.5 | 2.1 ± 0.9 | F = 0.4, $p$ = 0.650, P$\eta^2$ = 0.04 |
| **Omega-6 Fatty Acids (g·day$^{-1}$)** | 6.7 ± 2.6 | 5.2 ± 2.6 | 6.1 ± 3.4 | F = 1.2, $p$ = 0.327, P$\eta^2$ = 0.10 |
| **Trans Fats (g·day$^{-1}$)** | 1.0 ± 0.4 | 0.9 ± 0.5 | 0.8 ± 0.4 | F = 0.6, $p$ = 0.564, P$\eta^2$ = 0.05 |
| **Cholesterol (mg·day$^{-1}$)** | 369.1 ± 144.9 | 268.2 ± 104.6 | 312.5 ± 247.2 | F = 1.8, $p$ = 0.195, P$\eta^2$ = 0.13 |

$^a$ = different to before lockdown ($p \leq 0.05$)

$^b$ = different to during lockdown ($p \leq 0.05$). Mean ± standard deviation.

**Table 4. Minerals and trace elements recorded in the diets of highly trained adolescent swimmers before, during, and after the COVID-19 lockdown.**

| Nutrient | RNI [31] | Lockdown Period | | | Interaction effect |
|---|---|---|---|---|---|
| | | **Before** | **During** | **After** | |
| **Sodium (mg·day⁻¹)** | M: 1600 F: 1600 | 2754 ± 822 | 1784 ± 587[a] | 2848 ± 1256[b] | $F = 7.7, p = 0.003, P\eta^2 = 0.39$ |
| **Potassium (mg·day⁻¹)** | M: 3500 F: 3500 | 3471 ± 776 | 2615 ± 680[a] | 3052 ± 946 | $F = 3.6, p = 0.043, P\eta^2 = 0.23$ |
| **Chloride (mg·day⁻¹)** | M: 2500 F: 2500 | 4739 ± 1842 | 2719 ± 738[a] | 4413 ± 1541[b] | $F = 9.0, p = 0.001, P\eta^2 = 0.43$ |
| **Calcium (mg·day⁻¹)** | M: 1000 F: 800 | 1247 ± 371 | 858 ± 389[a] | 1219 ± 484[b] | $F = 7.6, p = 0.003, P\eta^2 = 0.39$ |
| **Phosphorus (mg·day⁻¹)** | M: 775 F: 625 | 1749 ± 294 | 1283 ± 374[a] | 1452 ± 466 | $F = 5.1, p = 0.014, P\eta^2 = 0.30$ |
| **Magnesium (mg·day⁻¹)** | M: 300 F: 300 | 336.0 ± 62.9 | 255.1 ± 83.4 | 432.3 ± 348.8 | $F = 2.4, p = 0.140, P\eta^2 = 0.17$ |
| **Iron (mg·day⁻¹)** | M: 11.3 F: 14.8 | 14.5 ± 3.2 | 10.1 ± 3.2[a] | 14.6 ± 6.0 | $F = 5.4, p = 0.012, P\eta^2 = 0.31$ |
| **Zinc (mg·day⁻¹)** | M: 9.5 F: 7.0 | 10.8 ± 2.3 | 8.8 ± 3.0 | 11.6 ± 6.1 | $F = 1.8, p = 0.183, P\eta^2 = 0.13$ |
| **Copper (mg·day⁻¹)** | M: 1.0 F: 1.0 | 1.4 ± 0.4 | 1.0 ± 0.4 | 1.5 ± 0.7 | $F = 3.1, p = 0.062, P\eta^2 = 0.21$ |
| **Manganese (mg·day⁻¹)** | - | 10.2 ± 20.0 | 2.8 ± 1.1 | 3.3 ± 1.1 | $F = 1.6, p = 0.226, P\eta^2 = 0.12$ |
| **Selenium (μg·day⁻¹)** | M: 70 F: 60 | 68.2 ± 15.7 | 49.4 ± 20.4[a] | 57.4 ± 28.2 | $F = 4.1, p = 0.030, P\eta^2 = 0.25$ |
| **Iodine (μg·day⁻¹)** | M: 140 F:140 | 187.3 ± 86.5 | 138.1 ± 88.6 | 159.8 ± 101.7 | $F = 1.2, p = 0.324, P\eta^2 = 0.09$ |

[a] = different to before lockdown ($p \leq 0.05$)

[b] = different to during lockdown ($p \leq 0.05$). RNI = British reference nutrient intakes for adolescents aged 15–18 years [31]. M = male RNI. F = female RNI.

Mean ± standard deviation.

($p = 0.027$, $g = 0.80$) fully increased compared to their lockdown intakes. A large effect size also suggests that iron increased after the lockdown period ($g = 0.91$), although this result did not reach statistical significance ($p = 0.071$). Whereas potassium ($p = 0.445$, $g = 0.51$), phosphorus ($p = 0.956$, $g = 0.39$), and selenium ($p = 0.947$, $g = 0.31$) did not increase again after lockdown, their level of intake did not differ compared to the intakes recording before lockdown (potassium: $p = 0.890$; phosphorus: $p = 0.309$; selenium: $p = 0.443$). Small-to-moderate effect between these two timepoints therefore suggest that these intakes may not have fully returned back to their normal levels (potassium: $g = 0.47$; phosphorus: $g = 0.74$; selenium: $g = 0.46$).

The majority of female participants consumed <14.8 mg·day⁻¹ of dietary iron before (75%, 13.9 ± 3.2 mg·day⁻¹), during (88%, 9.7 ± 3.0 mg·day⁻¹), and after lockdown (75%, 13.9 ± 3.2 mg·day⁻¹) (Fig 2). In addition, the percentage of females that consumed <800 mg·day⁻¹ of calcium increased from before (25%, 1327 ± 411 mg·day⁻¹) to during lockdown (50%, 920 ± 466 mg·day⁻¹), with 38% still intaking <800 mg·day⁻¹ once lockdown had ended (1099 ± 439 mg·day⁻¹). Further analysis also found that 50% (n = 4) of females were intaking <35 kcal·kg BM·day⁻¹ of energy, <14.8 mg·day⁻¹ of iron, and <800 mg·day⁻¹ of calcium at the same time during lockdown, compared to just 13% (n = 1) before and after. All five males consumed >11.3 mg·day⁻¹ of iron before (15.4 ± 3.3 mg·day⁻¹) and after lockdown (19.3 ± 3.6 mg·day⁻¹); however, three (60%) failed to achieve this threshold during the lockdown period (10.9 ± 3.7 mg·day⁻¹). Regarding calcium, only 60% (n = 3) of males consumed >1000 mg·day⁻¹ before lockdown (1120 ± 291 mg·day⁻¹), which further reduced to 20%

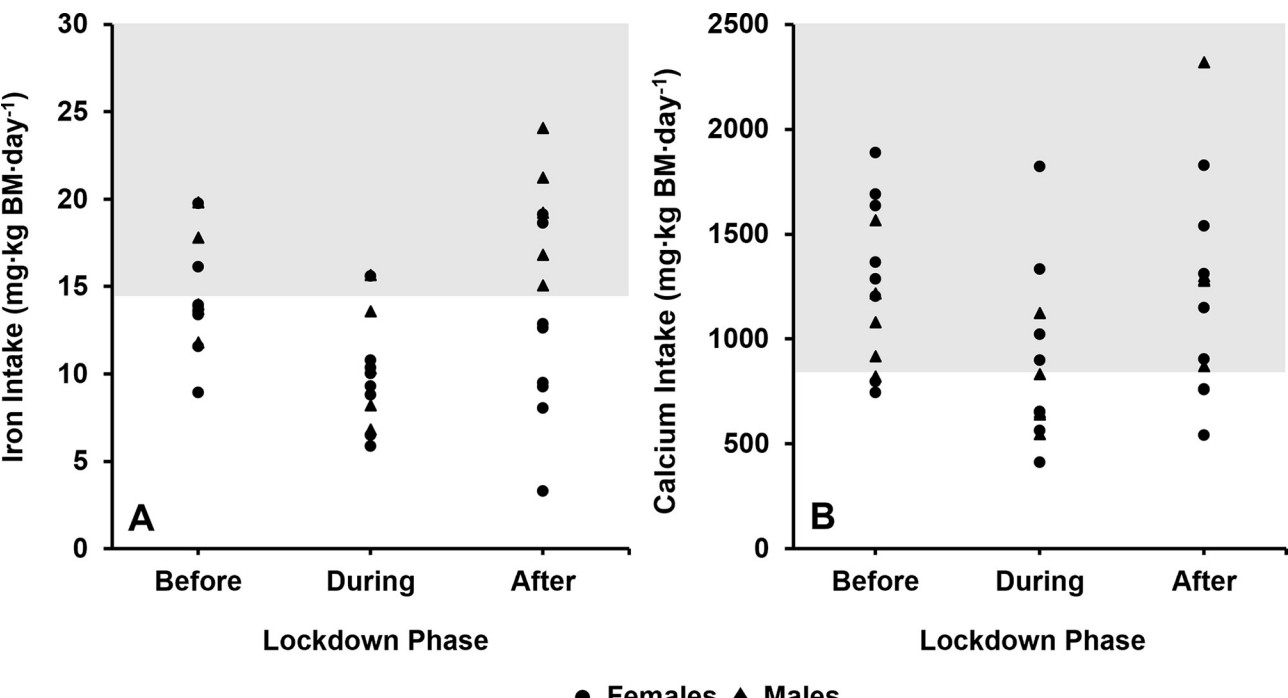

**Fig 2.** The A) iron and B) calcium intakes of each individual swimmer before, during, and after the COVID-19 lockdown. Grey shaded areas represent the reference nutrient intakes (RNI) recommended for females aged 15–18 years [31]. Note: iron RNI for males = 11.3 mg·day$^{-1}$, calcium RNI for males = 1000 mg·day$^{-1}$.

(n = 1) during lockdown (759 ± 229 mg·day$^{-1}$). Calcium intake increased amongst males after lockdown (80%, 1410 ± 539 mg·day$^{-1}$).

## Vitamins

Only riboflavin (F = 3.9, $p$ = 0.033, P$\eta^2$ = 0.25) and folate (F = 4.1, $p$ = 0.030, P$\eta^2$ = 0.25) were affected by the COVID-19 lockdown (Table 5); though, post-hoc analysis could not identify when these changes occurred. Based on moderate-to-large effect sizes, it is likely that these nutrients decreased in the diet during lockdown (before vs. during, riboflavin: $p$ = 0.102, $g$ = 0.77; folate: $p$ = 0.117, $g$ = 0.84), and increased again after the lockdown had ended (during vs. after, riboflavin: $p$ = 0.060, $g$ = 0.91; folate: $p$ = 0.067, $g$ = 0.98). No swimmer was intaking >10 μg·day$^{-1}$ vitamin D before lockdown (range: 1.8–8.9 μg·day$^{-1}$), but this increased to 15% (n = 2) during lockdown (range: 0.7–16.8 μg·day$^{-1}$), and 23% (n = 3) after the lockdown was lifted (range: 3.3–18.0 μg·day$^{-1}$).

## Caffeine

Caffeine intakes did not change over the course of the study (F = 1.9, $p$ = 0.189, P$\eta^2$ = 0.14), with negligible intakes reported before (0.1 ± 0.1 mg·kg BM·day$^{-1}$), during (0.1 ± 0.3 mg·kg BM·day$^{-1}$) and after (0.2 ± 0.5 mg·kg BM·day$^{-1}$) the lockdown.

## Discussion

This was the first study to have analysed the nutritional intakes of highly trained, British adolescent swimmers at three different time points: (a) before COVID-19, (b) during a national lockdown, and (c) during restricted training after lockdown. The key finding from this

**Table 5. Vitamin intakes recorded in the diets of highly trained adolescent swimmers before (January), during (April), and after (September) the 2020 COVID-19 lockdown.**

| Nutrient | RNI [31] | Lockdown Period | | | Interaction effect |
|---|---|---|---|---|---|
| | | Before | During | After | |
| Vitamin A (ret eq·day⁻¹) | M: 700 F: 600 | 758 ± 309 | 734 ± 363 | 1023 ± 634 | F = 2.0, $p$ = 0.157, Pη² = 0.14 |
| Vitamin D (µg·day⁻¹) | M: 10 F: 10 | 4.6 ± 2.3 | 4.0 ± 5.0 | 7.2 ± 5.0 | F = 2.0, $p$ = 0.155, Pη² = 0.14 |
| Vitamin E (mg·day⁻¹) | N/A | 9.2 ± 3.5 | 7.7 ± 2.9 | 11.4 ± 7.4 | F = 2.3, $p$ = 0.118, Pη² = 0.16 |
| Vitamin K (mg·day⁻¹) | N/A | 80.2 ± 58.8 | 84.9 ± 80.9 | 74.4 ± 52.1 | F = 0.1, $p$ = 0.876, Pη² = 0.01 |
| Thiamin (mg·day⁻¹) | M: 1.1 F:0.8 | 2.2 ± 0.4 | 1.8 ± 0.6 | 2.3 ± 0.8 | F = 2.1, $p$ = 0.145, Pη² = 0.15 |
| Riboflavin (mg·day⁻¹) | M: 1.3 F: 1.1 | 2.4 ± 0.7 | 1.8 ± 0.8 | 2.7 ± 1.1 | F = 3.9, $p$ = 0.033, Pη² = 0.25 |
| Niacin (mg·day⁻¹) | M: 18 F: 14 | 52.5 ± 9.7 | 49.5 ± 29.1 | 40.1 ± 13.8 | F = 1.4, $p$ = 0.270, Pη² = 0.10 |
| Pantothenic Acid (mg·day⁻¹) | N/A | 7.6 ± 1.7 | 6.9 ± 4.3 | 7.9 ± 3.5 | F = 0.3, $p$ = 0.645, Pη² = 0.03 |
| Vitamin B6 (mg·day⁻¹) | M: 1.5 F: 1.2 | 2.6 ± 0.7 | 2.4 ± 1.2 | 2.8 ± 0.9 | F = 0.6, $p$ = 0.499, Pη² = 0.06 |
| Folate (µg·day⁻¹) | M: 200 F: 200 | 310.5 ± 69.3 | 235.7 ± 99.6 | 355.8 ± 135.8 | F = 4.1, $p$ = 0.030, Pη² = 0.25 |
| Vitamin B12 (µg·day⁻¹) | M: 1.5 F: 1.5 | 7.3 ± 2.4 | 6.0 ± 3.5 | 6.6 ± 3.4 | F = 0.6, $p$ = 0.580, Pη² = 0.04 |
| Biotin (µg·day⁻¹) | N/A | 39.6 ± 7.1 | 28.8 ± 10.1 | 51.5 ± 35.2 | F = 3.6, $p$ = 0.076, Pη² = 0.23 |
| Vitamin C (mg·day⁻¹) | M: 40 F: 40 | 124.5 ± 78.0 | 104.8 ± 51.8 | 195.2 ± 163.9 | F = 3.3, $p$ = 0.085, Pη² = 0.22 |

[a] = different to before lockdown ($p \leq 0.05$)

[b] = different to after lockdown ($p \leq 0.05$).). RNI = British reference nutrient intakes for adolescents aged 15–18 years [31]. M = male RNI. F = female RNI.

Mean ± standard deviation.

investigation was that adolescent swimmers altered their nutrition based on their changing training demands, such that energy, CHO, protein, and fat intakes all decreased during lockdown, and increased again when swimming training returned. However, the reduced dietary intake during lockdown led to some micronutrients being consumed in amounts below the reference nutrient intakes (RNI) for British adolescents (aged 15–18 years [31]), which is concerning as athletes may have greater micronutrient needs. Furthermore, inter-individual variance increased for each nutrient after lockdown, suggesting that more severe overconsumption or underconsumption of nutrients might have been occurring on an individual basis. It is important to note that participants were receiving nutritional support throughout lockdown, therefore their ability to alter nutrient intakes may not be representative of other swimming cohorts without such support. Regardless, further research should identify effective and inexpensive methods to provide nutritional support to adolescent athletes to benefit their health and performance.

A 70% reduction in weekly training hours (-13 hours·week⁻¹) occurred in response to the COVID-19 lockdown, which suggests that there would have been large reductions in DEE at this time. To account for this reduced energy demand, the current adolescent swimmers reduced their consumption of dietary energy (-14.2 kcal·kg BM·day⁻¹), CHO (-1.9 g·kg BM·day⁻¹), protein (-0.6 g·kg BM·day⁻¹), and fat (-0.5 g·kg BM·day⁻¹). These reduced intakes were deemed to be appropriate since no body mass changes occurred one month into lockdown, demonstrating that energy balance was being achieved [32]. However, after four

months of lockdown, the swimmers returned to training with a mean body mass increase of 4.3 kg. Interestingly, this increase almost replicates previous findings in highly trained adolescent females (aged 17 ± 1 years), whereby 4.8 kg (4.3 kg of fat mass) was gained after a two-month detraining period [33]. It is therefore likely that the current swimmers also started to gain excess body fat when lockdown exceeded two months, either through increasing energy intake, or by further decreasing DEE [11, 33]. Alternatively, large body mass increases are common between the ages of 10–20 years, with height (e.g., +3 cm in the present study), skeletal muscle, and bone mass all rapidly increasing during adolescence [34]. With the lockdown causing prolonged reductions in training volume, it is also possible that this young cohort started to intake sufficient energy to promote optimal growth during this time [35]. In turn, a limitation is highlighted within this study since body composition could not be assessed to identify the source of increase. Future lockdown support could therefore include the analysis of simple, non-invasive techniques, such as waist-to-hip ratio, to appropriately infer the causes of body mass changes when less technical assessment methods are not possible [36].

Large dietary alterations were made by highly trained adolescents in response to changing training demands caused by the COVID-19 lockdown. This contrasts previous research in national and international-level swimmers [37, 38], although the current swimmers did receive sport nutrition support for the duration of the study. Despite dietary reductions during lockdown, group analyses showed that CHO, protein, and fat intakes were within the recommendations for swimmers at all time points [7, 10]. Energy intake was also >35 kcal·kg BM·day$^{-1}$ before and after lockdown, with the reduction to 31 kcal·kg BM·day$^{-1}$ during lockdown also appropriate based on the vast reduction in training hours. On an individual basis, however, each of these intakes increased in inter-individual variation after lockdown, signifying that more severe under- or over-consuming of energy/nutrients was taking place when training resumed (e.g., energy range before lockdown: 34–62 kcal·kg BM·day$^{-1}$ vs. after lockdown: 29–66 kcal·kg BM·day$^{-1}$; protein range before lockdown: 1.5–2.9 g·kg BM·day$^{-1}$ vs. after lockdown: 1.2–3.1 g·kg BM·day$^{-1}$). Indeed, other athletic populations reported a reduced diet quality (females: 64%, males: 47%) and training motivation (females: 40%, males: 48%) after five weeks of lockdown [8], therefore it is speculated that the current cohort also began to exhibit these behaviors during the later stages of lockdown (May-July). A combination of these factors is likely to reduce physical performance capacity [39], which could partly explain why 19% (5 of 26) of the possible cohort ended their swimming careers between March and September 2020 [40]. More interactive nutrition support (i.e., gamification strategies, social media use) may have therefore been required by some adolescent swimmers to incentivise adherence to optimal practices [41–43]. Given that novel coronavirus strains [44, 45] and injuries could lead to future periods away from sport for this cohort, swimming clubs should consider integrating these forms of nutrition education into day-to-day practice.

Though reducing energy intake was a necessary response to lockdown, swimmers should be aware that energy reductions also increase the risks of developing micronutrient deficiencies [22, 23]. This risk may be greater in athletic adolescents, who may require greater calcium (e.g., 1100–1500 mg [46, 47] and iron intakes (e.g., 22 mg·day$^{-1}$ for females [48]) than the general population to offset deficiency symptoms, such as illness, muscle fatigue, cognitive impairments, and osteoporosis [49, 50]. The current study adds to this concern by identifying reduced intakes for nine different vitamins and minerals during lockdown, including iron, calcium, potassium, and selenium, which all fell below the British reference nutrient intake (RNI) for adolescents (aged 15–18 years [31]). Furthermore, nether iron, selenium, nor potassium intakes increased again after lockdown, possibly due to a reduction in diet quality [8] and/or a displacement of nutrients by an increased sugar intake (+18%) [51]. More specifically, this research and others [52–54] reported suboptimal calcium (684–970 mg·day$^{-1}$) and iron intakes

(9.7–13.2 mg·day$^{-1}$) within female swimmers when energy intakes fall <40 kcal·kg BM·day$^{-1}$. Males are often considered to be less at risk since they routinely ingest higher energy intakes [7], although the current data suggests that this should be monitored on an individual basis. While this micronutrient data may be important, it is encouraged to be interpreted cautiously as the reliability of three-day food diaries to measure such values is currently unknown [25]. Nonetheless, these insights suggest that education and practical skills are of much needed attention within swimming clubs to ensure that adolescent swimmers can sustain diet quality year-round.

The retrospective nature of this study produced a number of limitations. Firstly, energy balance, and therefore risk of energy deficiency, was not included in analyses since a consistent DEE calculation method was not established during data collection. For example, the Harris-Benedict equation was used before lockdown using height, age, body mass, and daily activity [55], whereas metabolic equivalents (METs) were attempted during lockdown to account for the changes in daily routine [56, 57]. However, a caveat to the latter approach was that it involved an accurate recall of daily activity, which was not received by all participants. A second limitation was that engagement/attendance was not monitored for the virtual nutritional provisions. Anecdotal reports from the nutritionists suggest that engagement declined as lockdown extended, though it is unclear if this behaviour was demonstrated by the included participants. Finally, no health or performance measures were collected alongside dietary intakes, subsequently limiting any practical outcomes from being observed. These difficulties could be overcome in the future with the development of simple and validated athlete monitoring tools, which could allow dietary intake, activity, physical and psychological symptoms, exercise performance, and growth to be monitored remotely throughout the season. Nonetheless, the key purpose of this study was to identify any changes that occurred in the dietary intakes of highly trained adolescent swimmers during three different lockdown stages, to which this was achieved.

## Conclusion

With the aid of sport nutritionists, highly trained, UK-based adolescent swimmers modified their nutritional intakes during the COVID-19 pandemic, with dietary energy, CHO, protein, and fat intakes all decreasing during lockdown, and increasing again once swimming training returned. At the group level, macronutrient intakes met the recommendations for swimmers at all time points. However, more variable macronutrient intakes were identified after lockdown, which suggests that more swimmers could have been deviating away from the optimal recommendations. Micronutrient intakes followed a similar pattern and decreased during lockdown, with iron and calcium both falling below the RNI, but not fully increasing again for all swimmers after lockdown. Based on other athlete research during COVID-19, it is suspected that diet quality and motivation decreased in some swimmers when lockdown exceeded five weeks. Adolescent swimmers could therefore be at a greater risk of energy and nutrient deficiencies after lockdown, which could have long-term implications for health and performance. Future studies should investigate the effectiveness of interactive education and monitoring tools to maintain adolescents' adherence towards optimal nutrition practices in isolated periods away from training.

## Supporting information

**S1 Data. Anonymised data set.**
(XLSX)

## Acknowledgments

We gratefully acknowledge all participants who volunteered for the study. We would also like to thank Carl Grosvenor and Chris Littler from the City of Birmingham Swimming Club for facilitating the research process.

## Author Contributions

**Conceptualization:** Josh W. Newbury, Wee Lun Foo, Lewis A. Gough.

**Data curation:** Josh W. Newbury.

**Formal analysis:** Josh W. Newbury, Wee Lun Foo.

**Funding acquisition:** Lewis A. Gough.

**Investigation:** Josh W. Newbury, Wee Lun Foo.

**Methodology:** Josh W. Newbury, Wee Lun Foo, S. Andy Sparks, Lewis A. Gough.

**Project administration:** Josh W. Newbury.

**Validation:** Josh W. Newbury, S. Andy Sparks, Lewis A. Gough.

**Visualization:** Wee Lun Foo.

**Writing – original draft:** Josh W. Newbury, Wee Lun Foo.

**Writing – review & editing:** Matthew Cole, Adam L. Kelly, Richard J. Chessor, S. Andy Sparks, Mark A. Faghy, Hannah C. Gough, Lewis A. Gough.

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
