## [Decision Letter · Decision Letter 0]

21 Jan 2022

PONE-D-21-32306Nutritional intakes of highly trained adolescent swimmers before, during, and after a national lockdown in the COVID-19 pandemicPLOS ONE

Dear Dr. Sparks,

Thank you for submitting your manuscript to PLOS ONE. After careful consideration, we feel that it has merit but does not fully meet PLOS ONE’s publication criteria as it currently stands. Therefore, we invite you to submit a revised version of the manuscript that addresses the points raised during the review process.

We look forward to receiving your revised manuscript.

Kind regards,

Julien Louis, PhD

Academic Editor

PLOS ONE

Journal Requirements:

2. Please note that in order to use the direct billing option the corresponding author must be affiliated with the chosen institute. Please either amend your manuscript to change the affiliation or corresponding author, or email us at plosone@plos.org with a request to remove this option.

Reviewers' comments:

Reviewer's Responses to Questions

**Comments to the Author**

1. Is the manuscript technically sound, and do the data support the conclusions?

Reviewer #1: Yes

2. Has the statistical analysis been performed appropriately and rigorously? 

Reviewer #1: Yes

3. Have the authors made all data underlying the findings in their manuscript fully available?

Reviewer #1: Yes

4. Is the manuscript presented in an intelligible fashion and written in standard English?

Reviewer #1: Yes

5. Review Comments to the Author

Reviewer #1: In this study, the authors analyse the nutritional intakes of high-performance adolescent swimmers before, during and after lockdown. Using a longitudinal assessment of self-reported dietary intakes, they study the effect of lockdown restrictions on dietary practices of young swimmers benefiting from a virtual nutrition support. Based on this finding, they reported that adolescents swimmer seems to be able to appropriately adjusted their energy and macronutrient intakes to match their changing energy demands. The authors suggest promoting and studying the relevance of virtual nutrition support for athletes, especially in case of further periods of social isolation. Some important considerations are highlighted in this paper, but the importance of nutrition and the impact of the results found on the health and performance of young athletes should be more highlighted.

Introduction

l.77 The reference 17 is prior to COVID and concerns restrictive practices, the quote should not be at the end of the sentence to avoid confusion).

l.81 What is the reference for this value for 1,5g/kg/d? not the same in the paper from Shaw nor Desbrow.

Dietary references (for energy/CHO/fat) and RNI as part of the results and discussion should appear in the introduction.

Methods

What are your eligibility criteria? How did you determine the sample size? with a population of 13 athletes, I think you would have to justify a sample size at entry or a power test at exit.

Explain what sport science support is. The nutritional support discussed afterwards is not clearly defined

Explain that the nutritional values will be reported in absolute (EI, macro, minerals, vitamins) and relative to body weight (EI, macro, caffeine). Maybe even specify the reason?

Anthropometric Data

Specify the conditions of weighing (fasting, in underwear?) and the instructions given for home weighing.

Training Information

We don't know how you got the training characteristics data (activity diary filled out by the athletes?)

Statistical Analysis

For more fluidity in reading, separate the statistical analyses of the pilot study from other analyses.

Clarify what is done if normality is not respected?

Anthropometric Responses

I don't know if you have the data to calculate it ... but the biological gae estimated from the sitting height for example would be an interesting element to discuss the homogeneity of the group

Energy and Macronutrient Intakes

l.215 carbohydrate without -s. I even think it would be good to harmonize the rest of the manuscript with CHO.

l.218 Can we really say that there are even small differences with such significance? If all the subjects had the same tendency, argue in this sense. Especially since in the discussion it is specified that “After the lockdown, the swimmers consumed a diet that was no different in energy, macronutrient, or micronutrient content compared to before COVID-19”.

l.224 Quote here the reference 7 (I did not find on which reference Shaw et al bases itself to give this value?)

l.228 Do they consume more or less CHO than recommended?

l.231 It would be better to have a reference for excess and insufficient amount of protein intake

l.236 Reference for insufficient amount of fat intake

Tables 2, 3 and 4

Specify that the data are means and standard deviation

Specify that a = different to before (January) and b = different to during (April)

Carbohydrate and Lipid Components

l.253 participants demonstrated an altered amount of fiber, and monounsaturated fat compared to what? do they respect the recommendations

l. 294 put the reference (30)

l.299 the 13 before and 3 after swimmers are all female?

Discussion

l. 336 and 337 “this was the first study” twice, it's a bit repetitive

l.341 carbohydate without -s or CHO

l.343 to specify perhaps that the NRIs are not specific to the young athletes and quoted the citation reference

l.347 the difference in size between before and after is about 3 cm, right? (1.66 ± 0.09 vs. 1.69 ± 0.08 m; p = 0.002, g = 0.27).

Can we say that without having evaluated the impact of the nutritional program on this group of swimmers? the ability of some young swimmers to modify their diet to meet their needs may be enhanced by the nutrition program (hence the importance of explaining the objectives of the program beforehand)

It would be interesting to discuss the gender effect. Indeed, as Shaw et al. (2014) point out, males are more likely to match their food intake to their expenditure (at least in the sense of covering their needs when they increase).

l.363 or +3cm?

l.379 RNI 15-18 (l.343 RNI for adolescents aged 11-19years) Is this normal or a mistake?

l.387 Add a reference

l.396 Perhaps it would be necessary to moderate this information by adding "to our knowledge".

l.414 and in boys? if this is the case, it is possible to introduce the REDS ?

l.417 the paragraph on the method of assessing food intakes is either a methodological discussion point and should be at the beginning of the discussion or a limitation, to be discussed as such

What is the impact of variations in food intake on the health/performance of young athletes? Among other things, when they no longer respect the recommendations? Does the significant difference have a clinical meaning?

To emphasize the contribution of these results to the research in nutrition of athletes.

Conclusion

l.439 the previous studies on swimmers were not done under the same conditions (covid ...) quote the studies on other sports but done during the covid or specify the conditions?

l.439 There is a significant reduction in training (in hours and km/week) and a significant reduction in food intake but are these two reductions of the same order? is it possible to say that the athletes have approprietly adjuted their energy and macronutrient intakes?

6. PLOS authors have the option to publish the peer review history of their article (what does this mean?). If published, this will include your full peer review and any attached files.

Reviewer #1: No

---

## [Author Response · Author response to Decision Letter 0]

25 Feb 2022

We have attached a "response to reviewers" document to our resubmission.

---

## [Decision Letter · Decision Letter 1]

17 Mar 2022

Nutritional intakes of highly trained adolescent swimmers before, during, and after a national lockdown in the COVID-19 pandemic

PONE-D-21-32306R1

Dear Dr. Sparks,

We’re pleased to inform you that your manuscript has been judged scientifically suitable for publication and will be formally accepted for publication once it meets all outstanding technical requirements.

Kind regards,

Julien Louis, PhD

Academic Editor

PLOS ONE

Additional Editor Comments (optional):

The manuscript can be accepted pending a last minor correction as suggested by reviewer 1:

you specify in the method that the results are presented with mean and standard deviation in the tables, but you do not add the information correctly in the legend of the tables, I would put mean ± standard deviation rather than data ± standard deviation.

Reviewers' comments:

Reviewer's Responses to Questions

**Comments to the Author**

1. If the authors have adequately addressed your comments raised in a previous round of review and you feel that this manuscript is now acceptable for publication, you may indicate that here to bypass the “Comments to the Author” section, enter your conflict of interest statement in the “Confidential to Editor” section, and submit your "Accept" recommendation.

Reviewer #1: All comments have been addressed

2. Is the manuscript technically sound, and do the data support the conclusions?

Reviewer #1: Yes

3. Has the statistical analysis been performed appropriately and rigorously? 

Reviewer #1: Yes

4. Have the authors made all data underlying the findings in their manuscript fully available?

Reviewer #1: Yes

5. Is the manuscript presented in an intelligible fashion and written in standard English?

Reviewer #1: Yes

6. Review Comments to the Author

Reviewer #1: Thank you for your detailed answers and for taking into account my remarks. The nutritional intervention is more understandable and supports the discussion and conclusion. One last detail, you specify in the method that the results are presented with mean and standard deviation in the tables, but you do not add the information correctly in the legend of the tables, I would put mean ± standard deviation rather than data ± standard deviation. I look forward to hearing from you again.

7. PLOS authors have the option to publish the peer review history of their article (what does this mean?). If published, this will include your full peer review and any attached files.

Reviewer #1: No

---

## [Editor Report · Acceptance letter]

24 Mar 2022

PONE-D-21-32306R1 

Nutritional intakes of highly trained adolescent swimmers before, during, and after a national lockdown in the COVID-19 pandemic 

Dear Dr. Sparks:

I'm pleased to inform you that your manuscript has been deemed suitable for publication in PLOS ONE. Congratulations! Your manuscript is now with our production department. 

Kind regards, 

on behalf of

Dr. Julien Louis 

Academic Editor

PLOS ONE